# Plants Restoration Drives the Gobi Soil Microbial Diversity for Improving Soil Quality

**DOI:** 10.3390/plants13152159

**Published:** 2024-08-05

**Authors:** Lizhi Wang, Junyong Ma, Qifeng Wu, Yongchao Hu, Jinxiao Feng

**Affiliations:** 1Faculty of Hydraulic Engineering, Environment and Oceanography, Ludong University, Yantai 264025, China; yantailizhi@126.com; 2Institute of Field Water Conservancy, Soil and Fertilizer Research, Xinjiang Academy of Agricultural and Reclamation Sciences, Shihezi 832000, China; maymajy@163.com; 3Key Laboratory of Northwest Oasis Water-Saving Agriculture, Ministry of Agriculture and Rural Affairs, Shihezi 832000, China; 4Dongying Research Institute for Oceanography Development, Dongying 257091, China; hormole@163.com; 5Qingdao Institute of Technology, Qingdao 266300, China; fengjinxiao@qit.edu.cn

**Keywords:** Gobi Desert, ecological restoration, drought and salt stress, physicochemical properties, microbial diversity, soil quality improvement

## Abstract

Desertification and salt stress are major causes of terrestrial ecosystem loss worldwide, and the Gobi, representing a salt-stressed area in inland China, has a major impact on the ecosystems and biodiversity of its surrounding environment. The restoration of the Gobi Desert is an important way to control its expansion, but there are few studies on the evaluation of restoration. In this study, soils under different restoration scenarios, namely, soils in restored areas (R1, R2), semi-restored areas (SR1, SR2), and unrestored control areas (C1, C2), were used to investigate differences in microbial diversity and physicochemical properties. The results showed that the soil was mainly dominated by particles of 4–63 μm (26.45–37.94%) and >63 μm (57.95–72.87%). Across the different restoration levels, the soil pH (7.96–8.43) remained basically unchanged, salinity decreased from 9.23–2.26 to 0.24–0.25, and water content remained constant (10.98–12.27%) except for one restored sample in which it was higher (22.32%). The effective Al, Cu, and Zn in the soil increased, but only slightly. Total organic matter (TOM) decreased from 3.86–5.20% to 1.31–1.47%, and total organic nitrogen (TON) decreased from 0.03–0.06% to 0.01–0.02%, but the difference in total organic carbon (TOC) was not significant. High-throughput testing revealed that the bacterial population of the restored area was dominated by *A4b* (6.33–9.18%), *MND1* (4.94–7.39%), and *Vicinamibacteraceae* (7.04–7.39%). Regarding archaea, samples from the restored areas were dominated by *Marine Group II* (76.17–81.49%) and *Candidatus Nitrososphaera* (6.07–9.75%). PCoA showed that the different restoration levels were the main cause of the differences between the samples. Additionally, salinity was the dominant factor that induced this difference, but it was inhibited by the restoration and targeted enrichment of some of these functional genera. Desert restoration should therefore focus on conserving water rather than adding nutrients. Planting salt- and drought-tolerant vegetation will contribute to the initial restoration of the desert and the restoration of the microbiological content of the soil as it migrates over time, creating a cycle of elements. Restoration stimulates and enhances the microbial diversity of the soil via beneficial microorganisms.

## 1. Introduction

Land desertification is the main manifestation of the degradation of terrestrial ecosystem functions during urbanization and industrialization, leading to the destruction of habitat functions [1], the loss of soil and water conservation functions, and a reduction in species resources [2,3]. As a result of the generally low and inconsistent rainfall and inadequate surface runoff across the continent, the surrounding environment is highly susceptible to the effects of wind, sand, and drought, leading to the formation of new deserts and the occurrence of salt stress. However, the expansion of the Gobi Desert poses a significant threat to the human living environment, leading to water shortages and increasing the risk of pests and diseases [2,3]. Globally, 7.53% of land is desert, of which 16.03% is expanding. Vegetation diversity is a key element influencing desertification [4]. At the same time, the loss of habitat disrupts biodiversity and the normal migration of migratory birds, increases greenhouse gas emissions and the occurrence of dust and sand storms, and is detrimental to global ecosystem conservation and temperature control [5,6]. Xinjiang Uyghur Autonomous Region in China contains 380,000 square kilometers of desert, and the expansion of this geomorphology threatens the living environment of 21,815,800 people. In the last 20 years, desertification has been curbed through sustainable restoration, which has reduced the area of desertified land by 1956 square kilometers and improved living conditions for about 3,900,000 people in this area. Therefore, the restoration of the Gobi Desert through ecological techniques is an important means to curb the loss of inland soil functions in Western China. However, the assessment of restoration effects is often analyzed only through vegetation growth, and attention to microorganisms, which are the main carriers of soil functions, is still insufficient [7,8]. The microbial community, as an important participant in organic matter decomposition, element cycling, and vegetation growth, is at the core of soil ecological restoration. Meanwhile, the addition of some in situ functional strains to the restoration environment also contributes to the rapid recovery of soil functions and is one of the important methods of ecological restoration [9,10].

Desert restoration, particularly in the Gobi Desert, is a complex process that primarily focuses on re-establishing vegetation rather than merely decontaminating the soil. Unlike traditional soil remediation projects, the restoration of the Gobi Desert is centered on enhancing the soil’s water-holding capacity and fostering a robust microbial community. This approach is crucial for facilitating elemental cycling and metabolism in the high-salt-stress soil of the Gobi Desert, thereby ensuring the regrowth of native vegetation [9,10]. General restoration methods involve a combination of physical and biological interventions, but progress is often slow due to the arid nature of the Gobi Desert. The challenges include the instability of river flows, which limits the feasibility of large-scale water reservoirs for long-term supply [11]; the high costs and risks associated with large-scale physical restoration [12,13]; and the ecological impacts of rapid vegetation planting, which can lead to water scarcity and restoration failure in the short term [14]. Local experiences in the Gobi Desert have shown that initial restoration efforts typically involve planting shrubs such as *Hippophae rhamnoides* L. and *Haloxylon ammodendron* Bunge. These plants are chosen for their ability to stabilize sand and lock in moisture, inhibiting the spread of desertification and conserving water [15]. Based on the success of these initial efforts and local precipitation patterns, additional vegetation, including sand plants such as flowering stick, sand date palm, and white sand artemisia, are introduced to further enhance the ecosystem [15]. This process often requires partial remodeling of the terrain, fencing, and infrastructure development to support the newly established vegetation.

Changes in the physicochemical properties of soil can reflect the effect of restoration. For example, water content, pH, salinity, and nutrient levels are critical in the early stages of restoration. Monitoring these properties is essential for making timely adjustments to restoration strategies. This could involve targeted interventions such as replanting, irrigation, and soil renovation to enhance the restoration process [16]. Furthermore, changes in microbial diversity in the restored soil also play a significant role in identifying functional bacterial genera. This knowledge is important for future ecological restoration efforts, which must provide a sustainable source of microbiological agents without the risk of environmental pollution [16]. As a result, understanding the physicochemical properties and microbial diversity of the remediated soil in the Gobi Desert is fundamental to assessing the restoration scenario, conducting operations efficiently, and accelerating the restoration process. This comprehensive approach ensures that restoration efforts are not only effective but also sustainable in the long term.

This study was focused on the changes in soil properties under different restoration scenarios. We analyzed the relationship between physicochemical properties and soil microbial diversity to clarify the changing rules and the main bacterial genera. The findings have the potential to inform and improve restoration initiatives in other arid and saline regions around the world. By understanding how microbial communities contribute to soil fertility and agricultural productivity in such challenging environments, this research can serve as a blueprint for similar projects. The data and theoretical framework developed here can guide future applications of food waste in restoration efforts, ensuring that these interventions are more effective and sustainable.

## 2. Results

### 2.1. Physicochemical Properties

The pH of the soil in the three restoration scenarios was relatively close, at 7.96–8.43; the salinity showed a clear gradient, with remediated area (0.24–0.25 mg/kg) < semi-restored area (0.49–1.31 mg/kg) < non-restored area (2.26–9.23 mg/kg); and the water content of R1, was the highest at 22.32%, while the other soil samples’ water content was close at 10.98–12.66% (Table 1). The effective Fe content of C1 was the lowest, at 3.76 mg/kg, with the other samples having more closely grouped values of 8.63–15.18. The effective Mn content of SR1 was the lowest, at 1.97 mg/kg, and that of C1 was the highest, at 4.46 mg/kg; the other samples had closer values, at 3.15–3.60 mg/kg. The effective aluminum content was the highest in the remediated area, with 0.92–1.10 mg/kg, and lowest in C2, with 0.08 mg/kg. The other samples had closer values, at 0.32–0.56 mg/kg. Regarding the effective Cu content, the gradient was as follows: restored area (0.51–0.54 mg/kg) > semi-restored (0.44–0.45 mg/kg) > unrestored (0.20–0.35 mg/kg) (Table 1). As for the effective zinc content, R1 had the highest value, with 0.83 mg/kg, while the other samples had closer values (0.33–0.45 mg/kg). In terms of particle size, soil particles larger than 63 μm dominated, accounting for more than 57.95%. In terms of total organic matter (TOM), the gradient was as follows: restored area (1.31–1.47 mg/kg) < semi-restored (2.49–3.17 mg/kg) < unrestored (3.86–5.20 mg/kg); there was no significant difference in total organic nitrogen (TON) or total organic carbon (TOC) among the samples (Table 1). The above results indicate that the restoration effectively reduced the salt content in the soil; reduced the organic matter content in the soil; improved the water-holding capacity; and partially increased the content of active iron, copper, and zinc.

### 2.2. Bacterial Diversity

As shown in Figure 1, samples from the restored area were mainly dominated by *A4b* (6.33–9.18%), *MND1* (4.94–7.39%), and *Vicinamibacteraceae* (7.04–7.39%). Among them, *Cryobacterium* had the highest relative abundance, with 11.33% in R2 samples, followed by *Pseudarthrobacter*, with a relative abundance of 7.88%. There were differences between samples in the semi-restored area, with SR1 dominated by *Bacillus* (21.37%), *Jeotgalibacillus* (13.43%), and *Salimicrobium* (6.18%), while SR2 was dominated by *terrestrial group S0134* (10.09%), A4b (7.81%), and *subgroup 10* (7.19%). There were also large differences between samples in the unrestored area, but there was a clear head effect in both, which did not occur in the other areas. The C1 samples were dominated by *Alifodinibius* (29.04%), *Abidingimonas* (14.80%), *Nitrolancea* (8.22%), *Nitriliruptoraceae* (7.34%), *Salinisphaera* (6.47%), and *Halomonas* (6.28%). The C2 samples were dominated by *Halomonas* (27.32%), *BD2-11 terrestrial group* (9.77%), *Longispora* (9.12%), *Nitriliruptoraceae* (6.49%), and *LWQ8* (6.22%). The above results indicate that the soil microbial communities gradually grew more similar with the restoration process, but the microbial communities did not show obvious continuity between the restoration scenarios, and there were considerable differences between the communities. At the same time, only a small number of bacteria were able to survive under selective pressure in the unrestored soil, but the survival pressure or strategy varied, resulting in differences in the communities. Even the soil samples from partially remediated sites showed considerable variation in bacterial communities.

### 2.3. Archaeal Diversity

As shown in Figure 2, the samples in the restored areas were mainly dominated by *Marine Group II* (76.17–81.49%) and *Candidatus Nitrososphaera* (6.07–9.75%), with *Halogranum* and *Candidatus Nitrocosmicus* accounting for a certain percentage of samples in R1 and R2, respectively. The semi-restored areas were dominated by marine samples (6.07–9.75%). Samples from the semi-restored areas were dominated by *Marine Group II* (35.64–42.26%) and *Candidatus Nitrososphaera* (10.92–15.20%). Among these, *Haladaptatus* was also present in SR1 (8.27%), whereas *Candidatus Nitrocosmicus* (18.08%) and *Natronococcus* (8.19%) were more dominant in SR2. In the unrestored area, C1 was dominated by *Halostagnicola* (52.15%), *Haloterrigena* (11.76%), and *Halomicrobium* (8.65%), while C2 was dominated by *Candidatus Nitrosopumilus* (26.45%), *Marine Group II* (23.11%), *Aenigmarchaeota* (9.87%), and *Natronomonas* (7.21%). The above results indicate that targeted enrichment of archaea was effectively promoted in the restored soil and that the differences in the flora in the soil of the restored area were small, while there were large differences among the archaea in the soil of the unrestored areas.

### 2.4. Principal Coordinate Analysis for Bacteria and Archaea

A principal coordinate analysis (PCoA) (Figure 3) showed that the total explanation of the differences between the bacterial communities of the different samples on the first and second principal coordinate axes was 75.7%, with the differences due to the source mainly reflected on the first axis (54.3% explained). In addition, the differences between the restored area and the other two areas were partially reflected on the second axis. There was higher similarity between soil samples from the restored area than between other samples, and the results suggest that there are large differences in the bacterial communities in the soils of the restored, semi-restored, and unrestored areas of the Gobi Desert, especially in the restored soils, which are significantly different from the other two areas.

The PCoA (Figure 4) showed that the total explanation of the differences in archaeal communities between different samples on the first and second principal coordinate axes was 93.7%, in which the differences caused by the source were mainly reflected on the first coordinate axis (68.5%). The similarity between soil samples from the restored and semi-restored areas was higher than in the unrestored control area samples. The results indicate that there are large differences in the archaeal communities in the soil of the restored, semi-restored, and unrestored areas of the Gobi Desert, while some differences between the two samples of the soil from the unrestored areas are also worthy of attention.

### 2.5. Venn Analysis of Seasonal and Rooting Conditions

A Venn analysis (Figure 5) revealed that fewer than 35 bacterial OTUs were the same among all samples; the restored area had the most OTUs and shared the most with the semi-restored area, while the unrestored area had the fewest OTUs and shared more with the semi-restored area. These results indicate that the restoration of the Gobi Desert vegetation effectively promoted the increase in bacteria in the soil and that the semi-restored area lay between the restored and unrestored areas, partially overlapping with both.

Venn analysis revealed (Figure 6) that fewer than 13 OTUs of archaea were shared among all samples; the semi-restored area had the highest number of OTUs and the highest number of shared OTUs with the restored area, while the restored area had the lowest number of OTUs and the lowest number shared with the unrestored area. The above results show that the archaea increased and then decreased with the restoration of the Gobi Desert vegetation, probably because soil restoration promoted the increase in bacteria and occupied the ecological niche belonging to the archaea. Meanwhile, the soil in the semi-restored area still retained some of the characteristics of the unremediated soil, which provided a sufficient growth environment for the tolerant archaea.

### 2.6. Canonical Correlation Analysis of Bacteria and Archaea

CCAs based on bacterial communitiesshowed that *Halomonas*, *Alifodnibiu*, and the soil samples from the unrestored areas were positively correlated with and strongly influenced by soil salinity and moisture, and less so by effective Al (Figure 7). The genus *Bacillus* was positively correlated with and strongly influenced by only the SR1 samples and negatively correlated with the effective Al content (Table 2). The restored-area samples and SR2 samples were positively correlated with the effective Mn, effective Fe, and A4b bacterial species content (Table 2). The above results indicate that there are important differences in soil salinity and water content between the unrestored adjacent area and the other areas, and the effective content of elements other than Al tended to increase after restoration.

CCAs based on archaeal communities (Figure 8) showed that soil salinity, water content, effective Al content, and *Halostagnicola* were positively correlated with samples from the C1 area, while other unknown genera were positively correlated with samples from the C2 area (Table 3). Samples from restoration and semi-restored areas were positively correlated with *Marine Group II*, *Candidatus Nitrososphaera*, *Candidatus Nitrocosmicus*, effective Fe content, and effective Mn content. The above results indicate that soil salinity, moisture, and effective Al content were important characteristics of the unrestored area, but were limited by the presence of a large number of specific genera in the archaea that could not be screened, resulting in positive correlations only between C1 samples and salinity, moisture, and effective AL content in the archaeal CCA (Table 3).

## 3. Discussion

Samples from restored, semi-restored, and non-restored areas were selected in this study for physicochemical property analysis and high-throughput testing. We found that the salinity and organic matter content in the restored area decreased. This finding allows us to speculate that there are obvious microbial metabolic processes in the soil, while the organic matter in the soil of the non-remediated area is more likely to be evidence of the inability of the residual organic matter to be degraded. Semi-remediated areas are thought to have their own organic-matter-degrading bacteria, with the organic matter content falling between the levels of restored and unrestored areas [17].

The results of the high-throughput bacterial analysis show that the unidentified *A4b* gene fragment associated with nitrate-reducing bacteria [18], the ammonia-oxidizing bacterium genus *MND1*, and the complex organic substrate-utilizing microorganism *Vicinamibacteraceae* had a high relative abundance in both restoration zones [19,20]. The relative abundance of the genus *Lysobacter* [21], which is associated with pollution restoration in area R1 and vegetation eradication, was also high. Interestingly, *Cryobacterium* and *Pseudarthrobacter* [22,23], both of which had a high relative abundance in the R2 area, have some ability to tolerate low temperatures. However, they were found at very low levels in samples outside the restored area. This phenomenon may be due to the low winter temperature in Xinjiang Uyghur Autonomous Region in China. The susceptibility of those microorganisms to cold temperatures resulted in a slower recovery from low-temperature conditions. Meanwhile, vegetation growth after restoration provided metabolic opportunities for soil microorganisms and targeted enrichment of element-cycling bacterial genera. The relative abundance of bacteria varied considerably between the two samples from the semi-restored area. In the SR1 sample, the dominant microorganisms were the widespread genus *Bacillus* and the highly salt-tolerant genus *Jeotgalibacillus* [24,25]. This is consistent with the finding of higher soil salinity. However, the salt-tolerant genera S0134 terrestrial group and subgroup 10 dominated the SR2 sample; the relative abundance of *A4b*, representing the genus of nitrate-reducing bacteria, was also higher [26,27,28]. In contrast, salinity was still the main factor in the semi-restored area, but the bacterial community had already acquired some elemental cycling and metabolic capacity. Meanwhile, the collection sites of the semi-restored samples were farther apart than those of the restored samples. Hence, it is worthy of concern that the restoration has caused the enrichment of specific bacterial genera resulting in the loss of germplasm resources [28]. The C1 samples were mainly dominated by salt-tolerant bacterial genera, whereas the salt-tolerant bacterial genera in C2 were dominated by *Halomonas* [29], the *terrestrial group BD2-11*, and *Longispora* [30,31]. Combining the Venn analysis, PCoA, and CCAs, the differences between the restored, semi-restored, and unrestored plots were clear, and the main reason for the differences was soil salinity. However, there were greater differences in bacterial species and relative abundance between the soils in the semi-rehabilitated and unrehabilitated areas. Consequently, up to 54.3% of the variation in the PCoA was explained by the source of the samples, and even the differences between the rehabilitated samples and the other samples had at least a small percentage of variation explained on the PCo2 axis. This also resulted in salinity not being a dominant factor among all samples in the CCA compared to other environmental factors.

As for the archaea, the salt-tolerant bacterial taxon *Marine Group II* was prevalent in all samples except sample C1 [32]. Its abundance is often elevated in river estuaries, and it may have the ability to degrade macromolecules, according to existing studies [33]. However, it is not possible to speculate on function of the archaea in each sample or even whether they belong to the same genus of metabolically functional bacteria. The ammonia-oxidizing bacteria *Candidatus Nitrososphaera* and *Candidatus Nitrocosmicus* were more abundant in the remediated and semi-remediated soil samples [34,35], while the relative abundance of the salt-tolerant bacterium *Halogranum* was also higher in R1 [36]. The salt-tolerant bacterium *Halostagnicola* was mainly found in C1 [37], as were *Haloterrigena* and *Halomicrobium* [38,39]. The ammonia-oxidizing genus *Candidatus Nitrosopumilus* was relatively abundant in C2 [40]. A comparison of the archaeal and bacterial high-throughput analyses showed that salinity was one of the main features throughout the study area. Vegetation rooting effectively alleviated bacterial growth limitation in the high-salt environment. However, the growth of archaea was relatively slow, and the restoration effect had not yet led to the establishment of an oasis covered by large areas of vegetation [41]. This resulted in relatively similar archaeal species and abundances and the absence of methanogenic bacteria, which are commonly found in high abundance in soils with high organic matter content. Based on these findings combined with the PCoAs, the differences between samples due to the source are the main reason. Furthermore, this also suggests that the restoration measures are the main reasons for the changes in community structure through the prevalence of salt-tolerant bacterial genera. But for the prevalence of the bacterial genus Reducibacter, this suggests that N cycling processes are abundant throughout the restored area and that nitrogen fertilization is not the central element limiting restoration in the Gobi Desert [26,42].

## 4. Materials and Methods

### 4.1. Study Area

The study area was the saline and alkaline cultivated land at Xinjiang Production and Construction Corps (37°15′29″ E, 79°20′0″ N). It is a typical oasis area with self-pressurized irrigation in the inland arid zone at the edge of the Taklamakan Desert. The average annual number of sandstorms and floating and sinking days is about 220 d. To measure the restoration effect, we planted maize in this area; the depth of groundwater during the crop growth period from May to October is 1.5 to 2.0 m. According to the actual restoration scenario, a total of six typical sample sites (Figure 9) were selected for in situ surface soil sampling, including the sample sites of the fully restored areas (R1, R2), the semi-restored areas (SR1, SR2), and the unrestored control areas (C1, C2).

### 4.2. Samples

For each sample, 200 g of soil was collected using a sterile resin sampler and mixed in a sterile self-sealing bag, and large particles such as rhizomes and stones were removed; the remaining soil was divided among six 1.5 mL and three 50 mL sterile centrifuge tubes, which were placed in a −20 °C freezer and quickly transported to the laboratory at the Institute of Field Water Conservancy, Soil and Fertilizer Research, where they were frozen and stored in −80 °C and −20 °C freezers for microbiological and physicochemical property testing, respectively.

### 4.3. Determination of Properties of the Samples

pH and salinity: A mass of 10.0 g of each in situ sample was weighed into a 50 mL centrifuge tube and shaken with 1:5 ultrapure water for 30 min, after which the samples were centrifuged and measured with pH and salinity meters, respectively.

Soil moisture content: A mass of 20.0 g of each in situ sample was weighed into an aluminum drying dish and placed in a hot-air drying oven with the lid open to dry for 24 h. The lid was then left open to allow the samples to dry for another 12 h, after which they were weighed, and the two results were compared; if the results showed a decrease, the sample was dried for a further 12 h and weighed until the results no longer changed.

Detection of active metals: A volume of 20 mL of DTPA-CaCl2-TEA buffer solution (pH 7.3) was used as an extractant, and 10 g of air-dried soil was shaken in this solution for 2 h; the soil was then removed by filtration, and the level of active Cu in the filtrate was determined via inductively coupled plasma emission spectroscopy (ICP-OES), along with Fe, Mn, Zn, and Al [43].

Soil particle size and organic matter content tests: A small portion of each sample was used for grain-size analysis, and another portion was freeze-dried, homogenized, and ground in an agate mortar for analyses of TN, TOC, δ^13^C, and δ^15^N. The samples’ grain size was determined using a laser particle size analyzer (Mastersizer 2000, Malven Instruments Ltd., Malvern, UK) capable of analyzing particle sizes between 0.02 µm and 2000 µm after removing OM and carbonates with 15% H_2_O_2_ and 4 mol L^−1^ HCl [44]. The particle size data were classified as follows: <4 µm, clay; 4–63 µm, silt; and >63 µm, sand. Approximately 150 mg of each sample was weighed into a 5 × 8 mm tin capsule for the measurement of total carbon and TN via high-temperature combustion on an Elementar vario MACRO cube CHNS analyzer (UNSW Mark Wainwright Analytical Centre, Kensington, Australia), using HCl to remove inorganic carbon from samples.

High-throughput microbial screening: According to the standard protocol, sample DNA was extracted using the FastDNATM Spin Kit for Soil (MP Biomedicals, Santa Ana, CA, USA). A PCR was performed using archaeal primers AR109F/AR915R and bacterial primers BA27F/BA907R, targeting the V4–V5 region of 16S rRNA [45]. The reaction parameters were as follows: pre-denaturation at 94 °C for 2 min; 94 °C denaturation for 30 s; annealing at 55 °C for 30 s; and extension at 72 °C for 1 min. The reaction lasted for 25 cycles. PCR products were detected via gel electrophoresis and then cut and purified for high-throughput sequencing. The library was prepared with a TruSeq Nano DNA LT Library Prep Kit from Illumina (San Diego, CA, USA). The End Repair Mix2 kit was used to excise the base protruding from the 5′ end of DNA, complete the 3′ end, and add a phosphate group to the 5′ end. The NGS platform Illumina was used for sequencing, and the specific sequencing instrument was novaSeq. The original sequencing data were deposited in the National Center for Biotechnology Information (NCBI) database with the accession numbers SAMN41636963, SAMN41636964, SAMN41636965, SAMN41636966, SAMN41636967, SAMN41636968, SAMN41637197, SAMN41637198, SAMN41637199, SAMN41637200, SAMN41637201, and SAMN41637202.

The Quantitative Insights Into Microbial Ecology (QIIME) version 1.7.0 pipeline (http://www.Qiime.org) was used to process raw sequencing data with the default parameters [46]. Briefly, the representative sequences from each OTU were defined at a 97% identity threshold level, after which chimeric and low-quality reads were removed. Using the Ribosomal Database Project (RDP) classifier [47], the taxonomic classification of each OTU was assigned. The average relative abundance (%) of the predominant genus-level taxonomy in each sample was assessed by comparing the assigned sequence number of a particular taxon to the total obtained sequence number. To clarify microbial community differences, a PCoA and a Venn analysis were performed by R program (http://www.r-project.org) with the vegan package [45].

## 5. Conclusions

In summary, salinity was responsible for the major differences in bacterial communities between soil samples across restoration scenarios, and salt tolerance was an important characteristic of bacteria in semi-restored and unrestored environments, but there were differences between salt-tolerant genera. Restoration effectively mitigated the inhibitory effect of salinity and enriched the *Lysobacter*, a functional genus involved in vegetation removal, but had a limited effect on soil organic matter accumulation, which was insufficient to completely alter the archaeal community structure. Thus, the complete restoration process, which begins with the fixed suppression of the desert and ends with the planting of large quantities of vegetation, can be applied to the management of other desert environments. There is a clear process of targeted enrichment of soil microorganisms in this process, which offers the possibility of vegetation diversification.

In this study, laboratory analyses of the physicochemical properties and microbial diversity of soils at different stages of restoration showed that ecological restoration significantly reduced soil salinity, improved water holding capacity, and promoted soil fertility through an increase in beneficial microorganisms. In particular, improvements in soil microbial community structure were strongly correlated with restoration treatments, with increases in salt-tolerant and ammonia-oxidizing bacteria being critical for soil health. These findings provide strategies for ecological restoration in other arid and saline regions of the world, emphasizing the importance of water management and vegetation selection during restoration to achieve rapid recovery of soil function and long-term ecosystem stability. In addition, the data and theoretical framework of this study provide a scientific basis for the use of organic matter, such as food waste, for ecological restoration, ensuring the effectiveness and sustainability of these interventions and contributing to global progress in ecological conservation and land sustainability management.

## Figures and Tables

**Figure 1 plants-13-02159-f001:**
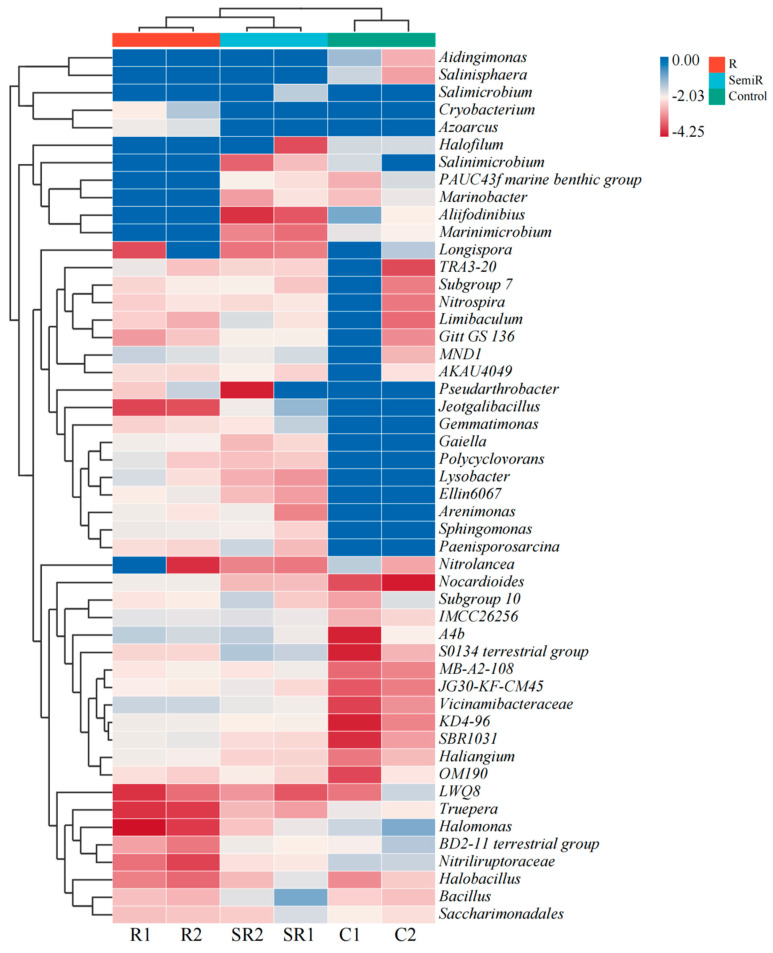
High-throughput bacterial assays of samples from the study areas. R1, R2: restored areas; SR1, SR2: semi-restored areas; C1, C2: unrestored control areas in the Xinjiang Gobi. R: restored area; SR: semi-restored area; C: unrestored control area in the Xinjiang Gobi. In the figure, the samples are UPGMA-clustered according to the Euclidean distances between the species. Species are clustered via UPGMA clustering. The data in the figure are shown on a logarithmic scale.

**Figure 2 plants-13-02159-f002:**
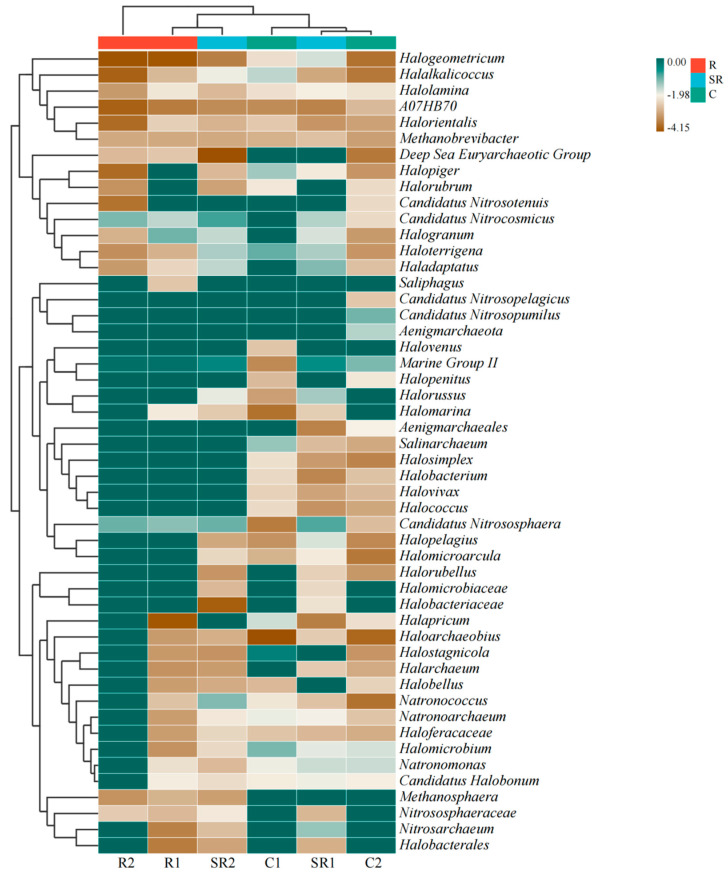
High-throughput assays of archaea in the study area. R1, R2: restored areas; SR1, SR2: semi-restored areas; C1, C2: unrestored control areas in the Xinjiang Gobi. R: restored area; SR: semi-restored area; C: unrestored control area in the Xinjiang Gobi. In the figure, the samples are UPGMA-clustered according to the Euclidean distances between the species. Species are clustered via UPGMA clustering. The data in the figure are shown on a logarithmic scale.

**Figure 3 plants-13-02159-f003:**
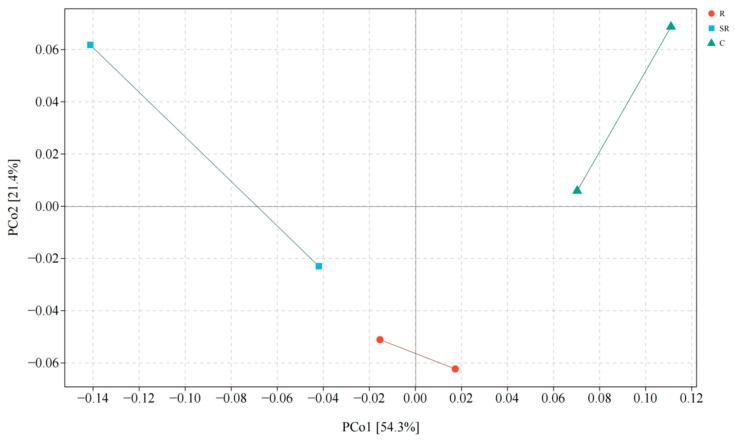
Principal coordinate analysis of bacteria. R: restored area; SR: semi-restored area; C: unrestored control area in the Xinjiang Gobi.

**Figure 4 plants-13-02159-f004:**
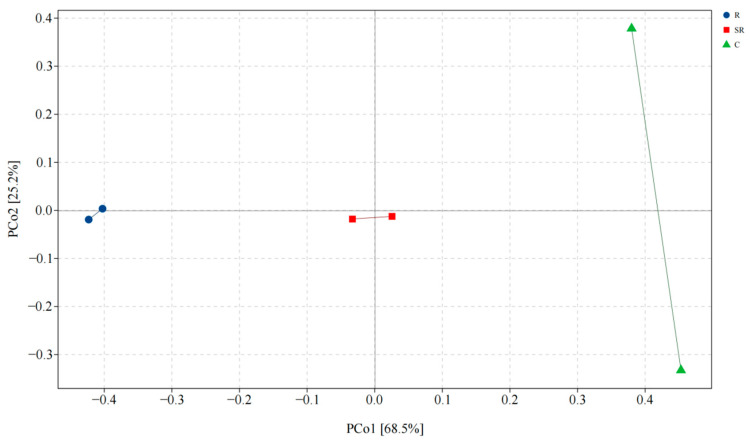
Principal coordinate analysis of archaea. R: restored area; SR: semi-restored area; C: unrestored control area in the Xinjiang Gobi.

**Figure 5 plants-13-02159-f005:**
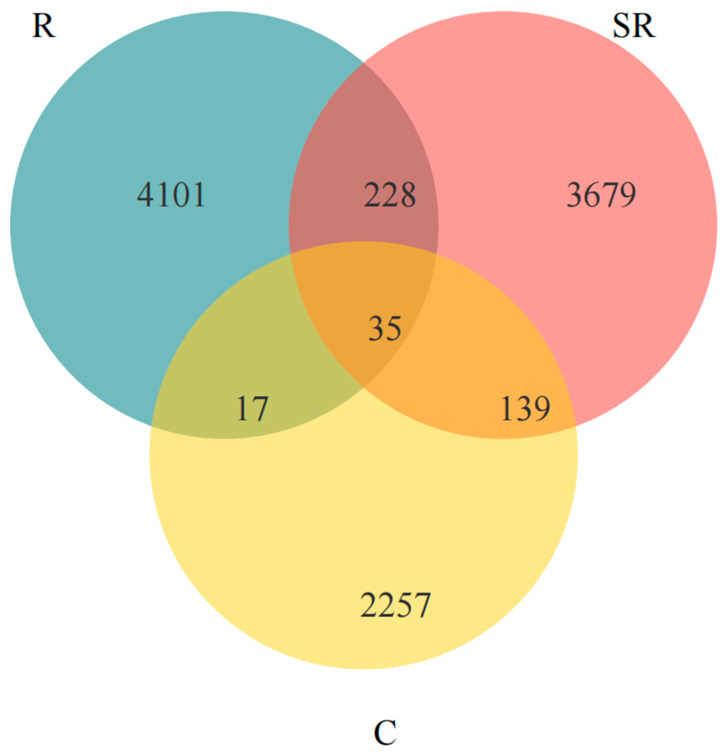
A Venn analysis based on different restoration statuses revealed the number of bacterial OTUs in each. R: restored area; SR: semi-restored area; C: unrestored control area in the Xinjiang Gobi.

**Figure 6 plants-13-02159-f006:**
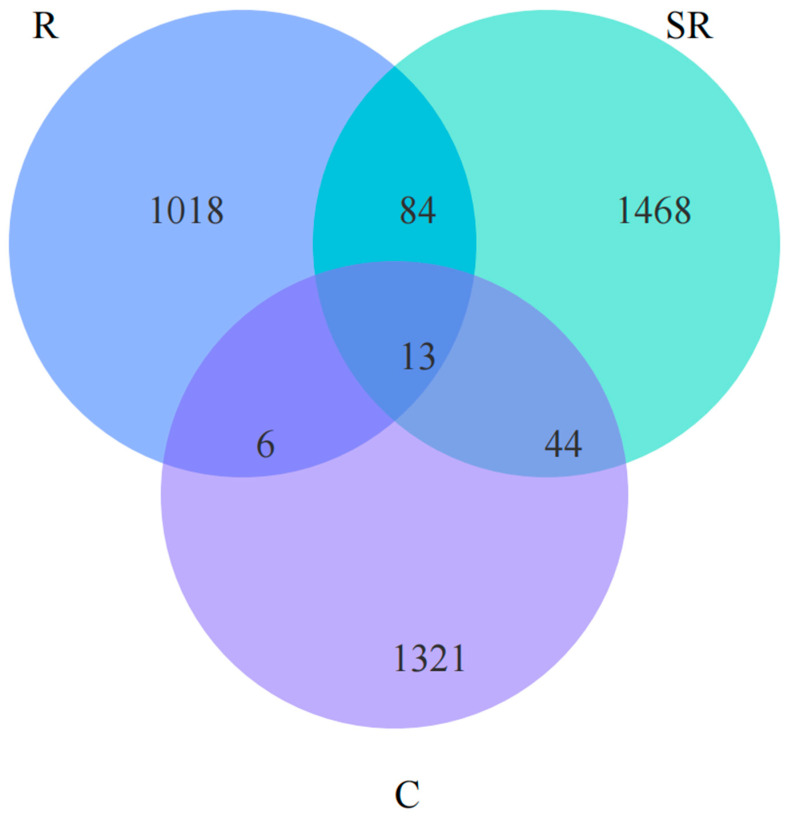
A Venn analysis based on different restoration statuses revealed the number of archaeal OTUs in each. R: restored area; SR: semi-restored area; C: unrestored control area in the Xinjiang Gobi.

**Figure 7 plants-13-02159-f007:**
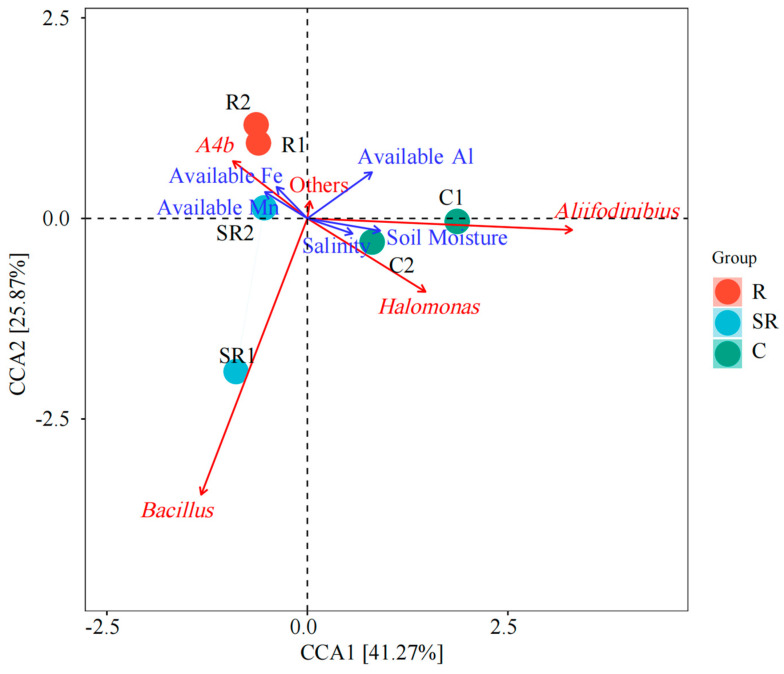
CCAs based on bacterial communities in samples. R: restored area; SR: semi-restored area; C: unrestored control area in the Xinjiang Gobi.

**Figure 8 plants-13-02159-f008:**
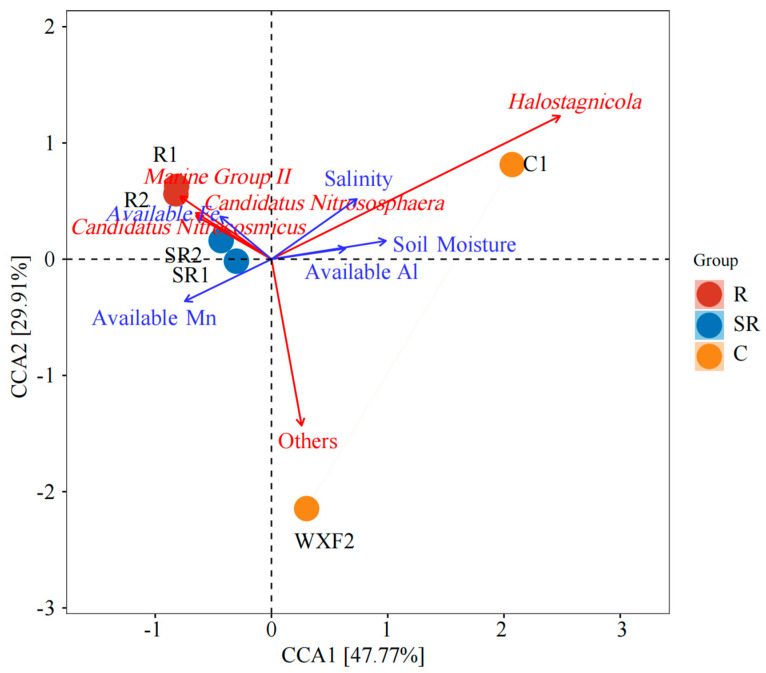
CCAs based on archaeal communities in samples. R: restored area; SR: semi-restored area; C: unrestored control areas in the Xinjiang Gobi.

**Figure 9 plants-13-02159-f009:**
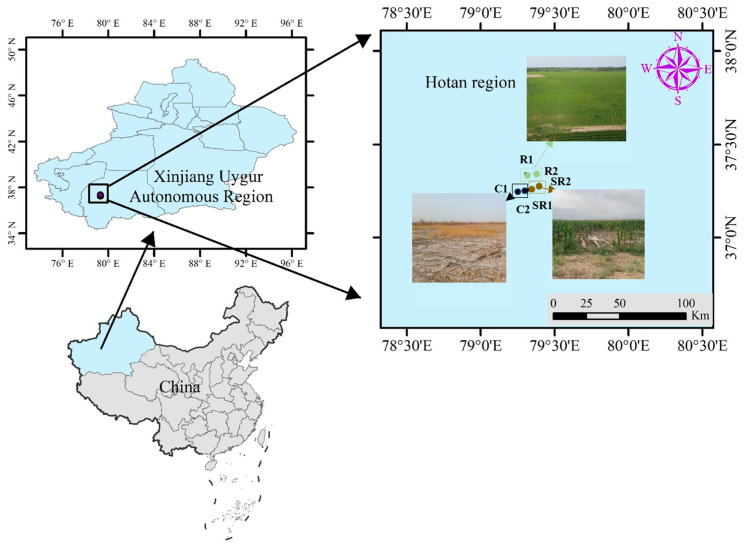
The study area: the saline and alkaline cultivated land at Xinjiang Production and Construction Corps. The black dots denote the stations in unrestored control areas (C1, C2); the brownish dots denote the stations in the semi-restored areas (SR1, SR2); the brownish dots denote the stations in the fully restored areas (R1, R2). The bule was represent the Xinjiang Uighur Autonomous Region.

**Table 1 plants-13-02159-t001:** Physicochemical properties of soil samples. R1, R2: restored areas; SR1, SR2: semi-restored areas; C1, C2: unrestored control areas in the Xinjiang Gobi.

	pH	Salinity	Water Content, %	Effective Fe, mg/kg	Effective Mn, mg/kg	Effective Al, mg/kg	Effective Cu, mg/kg	Effective Zn, mg/kg
R1	8.14	0.24	22.32	15.18	3.50	1.10	0.54	0.83
R2	8.04	0.25	12.27	10.12	3.28	0.92	0.51	0.43
SR1	8.18	1.31	12.66	8.63	1.97	0.32	0.45	0.37
SR2	7.96	0.49	12.60	9.60	3.15	0.40	0.44	0.33
C1	8.43	9.23	11.60	3.76	4.46	0.56	0.20	0.45
C2	7.98	2.26	10.98	12.74	3.60	0.08	0.35	0.33
	**Soil Particle Size**							
	**0–4 μm**		**4–63 μm**	**>63 μm**		**TOM%**	**TON%**	**TOC%**
R1	3.71		35.23	61.06		1.47	0.01	2.04
R2	2.77		30.21	67.02		1.31	0.02	1.99
SR1	4.11		37.94	57.95		3.17	0.03	2.12
SR2	2.93		33.77	63.30		2.49	0.04	2.32
C1	0.68		26.45	72.87		3.86	0.06	2.27
C2	2.06		33.49	64.45		5.20	0.03	2.45

**Table 2 plants-13-02159-t002:** Pearson correlation between different environmental factors based on bacterial communities in samples. ^a^ Pearson correlation rate; ^b^ *p*-value Pearson correlation confidence.

	R ^a^	P ^b^
Salinity	0.81	0.15
Soil Moisture	0.99	0.01
Available Fe	0.33	0.35
Available Mn	0.69	0.20
Available Al	0.41	0.52
Available Cu	0.66	0.14
Available Zn	0.99	0.01
Grain Size 0–4 μm	0.27	0.43
Grain Size 4–63 μm	0.74	0.16
Grain Size > 63 μm	0.54	0.35
TOM	0.58	0.33
TON	0.96	0.00
TOC	0.84	0.10

**Table 3 plants-13-02159-t003:** Pearson correlation between different environmental factors based on archaeal communities in samples. ^a^ Pearson correlation rate; ^b^ *p*-value Pearson correlation confidence.

	R ^a^	P ^b^
Salinity	0.35	0.53
Soil Moisture	0.85	0.12
Available Fe	0.30	0.51
Available Mn	0.39	0.48
Available Al	0.98	0.00
Available Cu	0.61	0.25
Available Zn	0.96	0.01
Grain Size 0–4 μm	0.27	0.57
Grain Size 4–63 μm	0.92	0.02
Grain Size > 63 μm	0.77	0.12
TOM	0.81	0.08
TON	0.76	0.12
TOC	0.73	0.15

## Data Availability

Data are contained within the article.

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
