# Peer review of "Plants Restoration Drives the Gobi Soil Microbial Diversity for Improving Soil Quality"

_plants, 2024, doi:10.3390/plants13152159_

Round 1

Reviewer 1 Report

Comments and Suggestions for Authors

Abstract – it is very detailed in terms of results, likely too long. There should be some conclusion for the importance of these results for other region that might be facing similar challenges.

Introduction

a)      there are many sentences that need to be rewritten as they are confusing or nonsensical.

b)      There should be proper data demonstrating how much the Gobi desert has been expanding and the number of people and total area affected by its increase. Without this kind of information, one has no idea the scale of the problem claimed by authors (if there is any) and therefore readers cannot value the importance of this study.

c)      The structure of this section is quite disorganized. Authors can use the first paragraph to lay out the context and main goal but the rest should be organized from a more broad to specific. As such, describe the general methods and problems in desert restoration narrowing down to the local experiences.

d)      Considering that this is an international journal and Gobi desert has global importance, authors should elaborate on how this study and its results can help inform other restoration initiatives around the world.

Conclusions – again, this section is strictly limited to this study results and no indication whether or not any lessons can be learned from it and apply elsewhere.

L42-43 – what is the evidence for “increases the stability of the ecosystem”?

L44 – The whole argument is to control the expansion of Gobi desert averting land degradation. In the end authors inserted a novel objective: “increases the productivity of productive crops”. Apart form the redundancy productivity of productive, the restoration for crop production was not contextualized.

L50 – “At the stage”. Which stage are authors referring to?

L52 – Please explain in the manuscript the meaning of “deep interior”.

L56 – What is “normal human environment”?

L54-57 – This sentence requires rewriting as its confusing and makes no sense, using awkward terms in a nonsensical logic (e.g. normal human environment, …., as it destroys the natural environment…

L58-60 – several citations are required here.

L62 – Inland soils? From the central part of the Gobi desert? Clarification is necessary here.

L67 and 70-73 – authors are using remediation and restoration interchangeably which is not correct. “objective is to achieve vegetation regrowth”(L71) is not remediation. Terminology should be based on a internationally accepted publication and its reference included.

L76 – river instability? Authors are referring to water supply or to the unstable river bed/banks?

L85, 87 – include scientific names for buckthorn, pike, flowering stick etc.

L86 – effect of restoration or the results restoration? Later (L337) authors use rehabilitation. Those terms do not have the same meaning.

L101 – How does the inadequate soild water holding capacity affects plants transpiration? (evapotranspiration is the sum of transpiration and evaporation)

L119 – Germplasm does not seem to be the adequate term here.

Figure 2 – provide full information about the picture. It is a dendrogram – mention it; a dendrogram based on what – describe it, etc.

Figure 3 and 4 – Captions do not fully inform readers about what is being represented. “r”, ”sr”and “C” should be described in full. There is no purpose for these graphs in the current format as readers cannot get any information from them. Finally, the correct acronym is PcoA.

Section 2.4 (L202-216) -The complete results for the PCoA analyses must be included.

L245-269 – Authors must include the complete CCA results. Graphics do not inform about the amount of information explained by the factor tested.

L274 – the study (L272) does not speculate (L274). People do. Also, results from a scientific study can indicate, or authors propose that the results  suggest something but leading to speculation is not a valid use of scientific studies.

Figure 9 – It is not clear what exactly the map is depicting in blue, is it China? The map should be improved by showing China, the province in which the study was conducted, and in inlet showing the Gobi desert and the study area.

L337 – Authors do not indicate how system stability has been improved based on the results. The same with microbial function.

Comments on the Quality of English Language

English editing is necessary as it is not adequate in the current state.

Author Response

For research article

Response to Reviewer 1 Comments

1. Summary

Thank you very much for considering our manuscript for review process in Plants. Thank you again for taking such valuable time to review our manuscript. Thank you for your valuable suggestions, which are very important for our work. We have carefully made point-to-point changes according your suggestions. We also have carefully revised the Figures and checked the whole manuscript. Please find the detailed responses below and the corrections highlighted in the re-submitted files. Please let us know if there is anything unsatisfactory.

2. Questions for General Evaluation

Reviewer’s Evaluation

Response and Revisions

Does the introduction provide sufficient background and include all relevant references?

Yes/Can be improved/Must be improved/Not applicable

Are all the cited references relevant to the research?

Yes/Can be improved/Must be improved/Not applicable

Is the research design appropriate?

Yes/Can be improved/Must be improved/Not applicable

Are the methods adequately described?

Yes/Can be improved/Must be improved/Not applicable

Are the results clearly presented?

Yes/Can be improved/Must be improved/Not applicable

Are the conclusions supported by the results?

Yes/Can be improved/Must be improved/Not applicable

3. Point-by-point response to Comments and Suggestions for Authors

Comments 1: Abstract – it is very detailed in terms of results, likely too long. There should be some conclusion for the importance of these results for other region that might be facing similar challenges.

Response 1: Thank you for pointing this out. We agree with this comment. Therefore, we have revised the abstract in Line 27-37 and marked in red.

Comments 2: Introduction

a) there are many sentences that need to be rewritten as they are confusing or nonsensical.

Response 2: Thank you very much for your comments, we have revised some sentences in Line 43, 48-51, 54, 56, 60, 64-97.

Comments 3: b) There should be proper data demonstrating how much the Gobi desert has been expanding and the number of people and total area affected by its increase. Without this kind of information, one has no idea the scale of the problem claimed by authors (if there is any) and therefore readers cannot value the importance of this study.

Response 3: Thank you very much for your comments, we have added these information, Line 48-51.

Comments 4: c) The structure of this section is quite disorganized. Authors can use the first paragraph to lay out the context and main goal but the rest should be organized from a more broad to specific. As such, describe the general methods and problems in desert restoration narrowing down to the local experiences.

Response 4: Thank you very much for your comments, we have reorganized the introduction, Line 64-96.

Comments 5: c) Considering that this is an international journal and Gobi desert has global importance, authors should elaborate on how this study and its results can help inform other restoration initiatives around the world.

Response 5: Thank you very much for your comments, we have added these information, Line 101-107.

Comments 6: Conclusions – again, this section is strictly limited to this study results and no indication whether or not any lessons can be learned from it and apply elsewhere.

Response 6: Thank you very much for your comments, we have added these information, Line 400-404

Comments 7: L42-43 – what is the evidence for “increases the stability of the ecosystem”?

Response 7: Thank you very much for your comments, we have revised these sentences, Line 36-37.

Comments 8: L44 – The whole argument is to control the expansion of Gobi desert averting land degradation. In the end authors inserted a novel objective: “increases the productivity of productive crops”. Apart form the redundancy productivity of productive, the restoration for crop production was not contextualized.

Response 8: Thank you very much for your comments, we have revised these sentences, Line 36-37.

Comments 9: L50 – “At the stage”. Which stage are authors referring to?

Response 9: Thank you very much for your comments, we have revised the expression, Line 43.

Comments 10: L52 – Please explain in the manuscript the meaning of “deep interior”.

Response 10: Thank you very much for your comments, we have revised the expression, Line 45-46.

Comments 11: L56 – What is “normal human environment”?

Response 11: Thank you very much for your comments, we have revised the expression, Line 49.

Comments 12: L54-57 – This sentence requires rewriting as its confusing and makes no sense, using awkward terms in a nonsensical logic (e.g. normal human environment, …., as it destroys the natural environment…48-50.

Response 12: Thank you very much for your comments, we have revised the sentence, Line 48-50.

Comments 13: L58-60 – several citations are required here.

Response 13: Thank you very much for your comments, we have added these information, Line 55.

Comments 14: L62 – Inland soils? From the central part of the Gobi desert? Clarification is necessary here.

Response 14: Thank you very much for your comments, we have revised the expression, Line 56.

Comments 15: L67 and 70-73 – authors are using remediation and restoration interchangeably which is not correct. “objective is to achieve vegetation regrowth”(L71) is not remediation. Terminology should be based on a internationally accepted publication and its reference included. All such as 62

Response 15: Thank you very much for your comments, we have revised all the expression, such as Line 61.

Comments 16: L76 – river instability? Authors are referring to water supply or to the unstable river bed/banks?

Response 16: Thank you very much for your comments, we have revised the expression, Line 71.

Comments 17: L85, 87 – include scientific names for buckthorn, pike, flowering stick etc.

Response 17: Thank you very much for your comments, we have revised the expression, Line 78.

Comments 18: L86 – effect of restoration or the results restoration? Later (L337) authors use rehabilitation. Those terms do not have the same meaning.

Response 18: Thank you very much for your comments, we have revised the expression in Line 74, Line 321.

Comments 19: L101 – How does the inadequate soild water holding capacity affects plants transpiration? (evapotranspiration is the sum of transpiration and evaporation) 97-98

Response 19: Thank you very much for your comments, we think this expression little relevance to the aim of the article, so deleted the expression.

Comments 20: L119 – Germplasm does not seem to be the adequate term here.

Response 20: Thank you very much for your comments, we have revised the expression in 89-92.

Comments 21: Figure 2 – provide full information about the picture. It is a dendrogram – mention it; a dendrogram based on what – describe it, etc.

Response 21: Thank you very much for your comments, we have added the information, Line 155-157, 176-178.

Comments 22 Figure 3 and 4 – Captions do not fully inform readers about what is being represented. “r”, ”sr”and “C” should be described in full. There is no purpose for these graphs in the current format as readers cannot get any information from them. Finally, the correct acronym is PcoA.

Response 22: Thank you very much for your comments, we have added the expression, Line 198-199, 201-202.

Comments 23: Section 2.4 (L202-216) -The complete results for the PCoA analyses must be included.

Response 23: Thank you very much for your comments, we have added the results, such as 184-185, 192-194.

Comments 24: L245-269 – Authors must include the complete CCA results. Graphics do not inform about the amount of information explained by the factor tested.

Response 24: Thank you very much for your comments, we have added table 2,3 for Pearson correlation between different environmental factors.

Comments 25: L274 – the study (L272) does not speculate (L274). People do. Also, results from a scientific study can indicate, or authors propose that the results suggest something but leading to speculation is not a valid use of scientific studies.

Response 25: Thank you very much for your comments, we have deleted these expression.

Comments 26: Figure 9 – It is not clear what exactly the map is depicting in blue, is it China? The map should be improved by showing China, the province in which the study was conducted, and in inlet showing the Gobi desert and the study area.

Response 26: Thank you very much for your comments, we have revised the Figure 9.

Comments 27: L337 – Authors do not indicate how system stability has been improved based on the results. The same with microbial function.

Response 27: Thank you very much for your comments, we have deleted this expression.

4. Response to Comments on the Quality of English Language

Point 1: English editing is necessary as it is not adequate in the current state.

Response 1: We have done extensive English revisions by English language editing services of the MDPI, Author Services ID: english-83163.

5. Additional clarifications

Thanks again to the editor and reviewer for their help. If you have any queries, please don't hesitate to contact me at the address below.

Reviewer 2 Report

Comments and Suggestions for Authors

The article presents a comprehensive analysis of soil properties and microbial diversity in several restoration options - fully restored, semi-restored and non-restored - a fact that allows a holistic understanding of these anthropic interventions to improve soil quality by restoring plants in the desert Gobi. This comparative approach allows better understanding of the specific benefits and limitations of different levels of restoration. The relevance of the study is highlighted against the background of the increasingly sustained concerns of specialists regarding land degradation through desertification processes.

The use of high-throughput sequencing techniques provides detailed information about the microbial communities present in soil, which improves the accuracy and depth of microbial analysis.

It is a well conducted, well analyzed and well written study. I recommend publication on the condition that the Material and Methods section be moved after the Introduction.

To increase the applicability of the research results, I recommend the authors carry out future studies on the causes of the variability of microbial communities in semi-restored areas. Also, the study of fungi could enrich the information regarding the microbial ecosystem in the soil.

Author Response

For research article

Response to Reviewer 2 Comments

1. Summary

Thank you very much for considering our manuscript for review process in Plants. Thank you again for taking such valuable time to review our manuscript. Thank you for your valuable suggestions, which are very important for our work. We have carefully made point-to-point changes according your suggestions. Please find the detailed responses below and the corrections highlighted in the re-submitted files. Please let us know if there is anything unsatisfactory.

2. Questions for General Evaluation

Reviewer’s Evaluation

Response and Revisions

Does the introduction provide sufficient background and include all relevant references?

Yes/Can be improved/Must be improved/Not applicable

Are all the cited references relevant to the research?

Yes/Can be improved/Must be improved/Not applicable

Is the research design appropriate?

Yes/Can be improved/Must be improved/Not applicable

Are the methods adequately described?

Yes/Can be improved/Must be improved/Not applicable

Are the results clearly presented?

Yes/Can be improved/Must be improved/Not applicable

Are the conclusions supported by the results?

Yes/Can be improved/Must be improved/Not applicable

3. Point-by-point response to Comments and Suggestions for Authors

Comments 1: The article presents a comprehensive analysis of soil properties and microbial diversity in several restoration options - fully restored, semi-restored and non-restored - a fact that allows a holistic understanding of these anthropic interventions to improve soil quality by restoring plants in the desert Gobi. This comparative approach allows better understanding of the specific benefits and limitations of different levels of restoration. The relevance of the study is highlighted against the background of the increasingly sustained concerns of specialists regarding land degradation through desertification processes.

The use of high-throughput sequencing techniques provides detailed information about the microbial communities present in soil, which improves the accuracy and depth of microbial analysis.

It is a well conducted, well analyzed and well written study. I recommend publication on the condition that the Material and Methods section be moved after the Introduction.

Response 1: Thank you for your comments. Normally journals are published with the material and methods placed after the Introduction. However, the format in Plant requires that the material and methods be placed before the conclusion and after the discussion. Sorry, but we can only typeset to the Plants specifications.

Comments 2: To increase the applicability of the research results, I recommend the authors carry out future studies on the causes of the variability of microbial communities in semi-restored areas. Also, the study of fungi could enrich the information regarding the microbial ecosystem in the soil.

Response 2: Thank you very much for your comments, we will pay more attention on the causes of the variability of microbial communities in semi-restored areas in the future and added some information in Line 400-404 and marked in red.

4. Response to Comments on the Quality of English Language

Point 1:

Response 1: We have done extensive English revisions by English language editing services of the MDPI, Author Services ID: english-83163.

5. Additional clarifications

Thanks again to the editor and reviewer for their help. If you have any queries, please don't hesitate to contact me at the address below.

Round 2

Reviewer 1 Report

Comments and Suggestions for Authors

Manuscript has been improved.

The following comments were not directly addressed and should be amended.

There should be proper data demonstrating how much the Gobi desert has been expanding and the number of people and total area affected by its increase. Without this kind of information, one has no idea the scale of the problem claimed by authors (if there is any) and therefore readers cannot value the importance of this study.

c) Considering that this is an international journal and Gobi desert has global importance, authors should elaborate on how this study and its results can help inform other restoration initiatives around the world.

Although I pointed out to the lack on information on the caption of some figures authors should improved all of them.

Comments on the Quality of English Language

English editing is required as many sentences are awkwardly written.

Author Response

For research article

Response to Reviewer 1 Comments

1. Summary

Thank you very much for taking the time to re-review our manuscript. Thank you very much for giving the constructive comments for our papers. We have improved the manuscript exactly as your comments. Please find the detailed responses below and the corrections highlighted in the re-submitted files.

2. Questions for General Evaluation

Reviewer’s Evaluation

Response and Revisions

Does the introduction provide sufficient background and include all relevant references?

Yes/Can be improved/Must be improved/Not applicable

Are all the cited references relevant to the research?

Yes/Can be improved/Must be improved/Not applicable

Is the research design appropriate?

Yes/Can be improved/Must be improved/Not applicable

Are the methods adequately described?

Yes/Can be improved/Must be improved/Not applicable

Are the results clearly presented?

Yes/Can be improved/Must be improved/Not applicable

Are the conclusions supported by the results?

Yes/Can be improved/Must be improved/Not applicable

3. Point-by-point response to Comments and Suggestions for Authors

Comments 1: There should be proper data demonstrating how much the Gobi desert has been expanding and the number of people and total area affected by its increase. Without this kind of information, one has no idea the scale of the problem claimed by authors (if there is any) and therefore readers cannot value the importance of this study.

Response 1: Thank you for pointing this out. We have added those information in Line 54-59 and marked in red.

Comments 2: c) Considering that this is an international journal and Gobi desert has global importance, authors should elaborate on how this study and its results can help inform other restoration initiatives around the world.

Response 2: Thank you very much for your comments. we have added those information in Line 413-426.

Comments 3: Although I pointed out to the lack on information on the caption of some figures authors should improved all of them.

Response 3: Thank you very much for your comments. We have added those information for improving in Figure 1 in Line 157-158, 160-161; Figure 2 in Line 180-181, 183-184; Figure 5 in Line 218-219; Figure 6 in Line 231-232; Figure 7 in Line 244-245; Figure 8 in Line 260-261.

4. Response to Comments on the Quality of English Language

Point 1: English editing is required as many sentences are awkwardly written

Response 1: Thank you very much for your comments. The English editing was conducted by the MDPI Autor Service (Manuscript ID: english-83163). In particularly, we carefully modified some awkwardly sentences in Lines 19-25, 30-33, 61-63, 69-70, 88-93, 101-103, 267-270, 279-284, 287-292, 294-297, 301-306, 309-320, 322-330 and mark in red.

5. Additional clarifications

Thanks again to the editor and reviewer for their help. Please let us know if there is anything unsatisfactory.
